# CROSS-MODAL SEMANTIC ANCHORING: UNSUPERVISED CORRESPONDENCE LEARNING FOR AERIAL IMAGERY AND MAPS VIA MULTIMODAL LLMS

## ABSTRACT

Up-to-date maps are crucial for urban living, enabling navigation, planning, and decision-making. The increasing accessibility of aerial imagery provides a cost-effective solution for updating map semantics, particularly the representation of buildings, which reflects ongoing urban renewal through construction and demolition. However, aligning heterogeneous modalities—aerial images and maps—remains challenging due to the significant modality gap. While previous works focus on low-level visual feature matching, we argue that these methods ignore the semantic correspondence between the map and aerial imagery. Therefore, we propose U-CSA, an unsupervised cross-modal semantic anchoring framework powered by multimodal large language models (MLLMs). Unlike conventional contrastive pre-training approaches that rely on large paired datasets, U-CSA exploits the world knowledge and cross-modal reasoning capabilities of MLLMs to generate high-level semantic anchors—interpretable descriptions of salient geo-entities and spatial structures. These anchors provide a unified semantic space, guiding dual-branch image encoders to align visual features through anchored contrastive learning. The semantically enriched encoders are then incorporated into an adversarial matching network, where dynamically generated sample pairs enable fine-grained discrimination between matched and unmatched regions. Extensive experiments demonstrate the superiority of U-CSA over other state-of-the-art approaches.

## 1 INTRODUCTION

With rapid urbanization, maps play a crucial role by providing a referencing framework that links the physical world to virtual environments. To ensure their validity, maps need to be regularly updated as changes occur in the real world (Fesenmyer et al., 2025; Ding et al., 2020). However, due to the high costs associated with manual updates, maps are typically revised only in response to substantial changes. In this work, we address the fundamental challenge of updating buildings on maps, as urban renewal is primarily manifested in the construction and demolition of buildings (Cao et al., 2017). Traditionally, this task relies on surveying and mapping, which are both labor-intensive and expensive. The increasing availability of aerial imagery offers a cost-effective alternative, making it possible to leverage computer vision techniques to retrieve up-to-date geographic information (Zhan et al., 2017).

This study focuses on the core task of updating buildings in maps using aerial imagery. However, directly aligning or comparing these two data modalities—aerial images and maps—remains highly challenging. Due to significant differences in imaging mechanisms, sensor perspectives, and levels of information abstraction, there exists a notable modality gap between them: images are characterized by rich real textures, lighting variations, and fine details, whereas maps represent geographic entities in a highly abstract and symbolic manner. This gap severely limits the performance of existing matching methods that rely on low-level visual features such as color, texture, and edges—especially in complex urban environments (Deng et al., 2024). We argue that this task is feasible, as the semantic information contained in maps is inherently associated with that of aerial imagery. Human observers, for instance, can establish connections between images and maps through high-level semantic reasoning (e.g., recognizing this is a stadium located by the river). Therefore,

the core scientific question addressed in this work is: How to establish reliable and robust semantic correspondences for heterogeneous data modality?

This challenge has prompted us to redefine the map update task as: identifying semantic consistency between aerial images and maps. That is, given an image and a map, the model is expected to determine whether they are semantically consistent. In practice, we have found a key data imbalance: positive sample pairs (matching image-map pairs) representing "no change" or "minor changes" are readily available, particularly in regions where urbanization has stabilized. In contrast, negative sample pairs that capture significant changes are scarce and difficult to collect at scale. This imbalance naturally motivates an unsupervised learning paradigm, i.e., can we learn a powerful feature representation that bridges the modality gap using only the abundant positive sample pairs?

A closer examination of existing deep learning approaches highlights why such an unsupervised strategy is needed. First, although many current works claim to address "heterogeneous image" matching, they typically focus on remote sensing images captured by different sensors (e.g., optical versus Synthetic Aperture Radar (SAR) images) or remote sensing images of varying resolutions (Zhong et al., 2022; Sun et al., 2021; Dong et al., 2024). While these tasks are non-trivial, the modality gap involved is much narrower than that between real aerial images and highly abstract, symbolic maps. Second, many advanced matching models (Chen et al., 2024a; Bastani & Madden, 2021) rely heavily on supervised or semi-supervised learning paradigms, requiring large-scale annotated datasets comprising both matching and non-matching sample pairs. This requirement constitutes a key bottleneck in practice, as it ignores the realistic scenario where positive (matching) pairs are readily available, but negative (non-matching) pairs are difficult to obtain at scale. Therefore, there is an urgent need for a truly unsupervised learning framework that can effectively cope with the huge modality gap between aerial images and maps.

To address the above challenges, this paper proposes a novel framework termed "Unsupervised Cross-Modal Semantic Anchoring (U-CSA)". The core idea of U-CSA is to move beyond direct comparison of heterogeneous visual features and instead introduce an external knowledge source—a pre-trained multimodal large language model (MLLM). For each matching image-map pair, the MLLM generates high-level, shared semantic descriptions (e.g., a stadium by the river, a residential area surrounded by parks). These context-rich text descriptions generated by MLLM serve as the "semantic anchors" that bridge the substantial gap between the two modalities.

Within the U-CSA framework, we design a dual-branch encoder architecture inspired by contrastive learning paradigms such as CLIP (Radford et al., 2021). Unlike traditional methods, which directly pull matching images closer as positive samples, our approach implicitly and efficiently aligns aerial imagery and maps within a shared high-level semantic space constructed by the MLLM. This is accomplished by maximizing the mutual information between visual features of the image-map pair and those of their corresponding "semantic anchor" text, without relying on manual negative sampling. Finally, the encoder enhanced by semantic anchoring can learn features that are insensitive to modality gap and highly consistent with semantic content, thereby achieving accurate and robust image-map matching. Our main innovations are as follows:

- We abandon the traditional "image-image" contrastive learning approach and innovatively use "semantic anchor" texts generated by the MLLM as positive samples for visual features. This shifts the model's learning objective from pixel-level similarity to higher-dimensional semantic consistency.
- We introduce a novel dual-branch encoder architecture that effectively aligns aerial imagery and maps within a shared semantic space, leveraging the contextual knowledge provided by the MLLM.
- We introduce the use of MLLM as a knowledge bridge, facilitating the implicit alignment of the two visual modalities within a shared semantic space through mutual information maximization, without requiring manual negative sampling.

## 2 RELATED WORK

Recent research increasingly frames automatic map updating as an incremental refinement problem rather than full reconstruction from scratch. The MUNO21 dataset exemplifies this shift, supporting operations such as adding, removing, and modifying roads while preserving validated regions

(Bastani & Madden, 2021). Other works have focused on vector–image alignment and annotation correction, including deep learning methods for cadastral misalignment correction (Pisl et al., 2021), detection of missing buildings in OSM (Vargas-Muñoz et al., 2019). General vector-map alignment approaches based on optimization and self-supervision have also been proposed (Cherif et al., 2024). With the rise of foundation models (e.g., *SAM* (Kirillov et al., 2023)), zero-shot segmentation has enabled unsupervised discrepancy detection between optical imagery and OSM, scaling map maintenance without extensive supervision (Cherif et al., 2024). Meanwhile, comprehensive surveys and advanced methods on road extraction and vectorization provide essential methodological context (Mo et al., 2024; Sahu & Ohri, 2019). Collectively, these works establish a foundation for discriminative frameworks trained only on matching pairs by introducing strategies for alignment, error correction, and handling incomplete annotations.

Unsupervised change detection has also contributed significantly to this area, specifically in tackling the challenges of heterogeneous-sensor analysis, such as optical-SAR data comparison. Many approaches learn modality-invariant representations: for instance, structural relationship graphs with graph-convolutional autoencoders (SR-GCAE) capture structural similarity in multimodal pairs (Chen et al., 2022), while Markovian co-segmentation jointly infers change masks across heterogeneous modalities (Sun et al., 2021). Other methods follow a harmonize then compare strategy, using cross-modal translation (with change-aware priors to ignore changed regions (Luppino et al., 2021)) or specialized fusion networks (Wang et al., 2024) to achieve harmonization. Self-supervised contrastive learning is also widely applied: weighted contrastive losses with context-aware Transformers alleviate change leakage in SAR (Dong et al., 2024), radar–optical masked autoencoders learn transferable bimodal features (Fuller et al., 2023), and semantic-to-change triplets explicitly model temporal differences (Ding et al., 2025). While effective for cross-sensor change detection, these approaches are not designed for the map update problem, where the challenge lies in bridging the geometric and semantic gaps between maps and remote sensing imagery.

In contrast, we introduce the task of unsupervised image-map matching for automated map updating. Our model is trained exclusively on matched image-map pairs. At inference time, it produces a matching probability, thereby directly bridging the geometric and semantic gaps between maps and images. Unlike heterogeneous change detection methods, which aim for sensor-invariant representations across modalities such as SAR and optical imagery, our approach addresses the semantic alignment between texture-rich aerial images and abstract symbolic maps. This formulation provides a practical pathway for real-world map updating without reliance on large annotated change datasets or multi-sensor alignment pipelines.

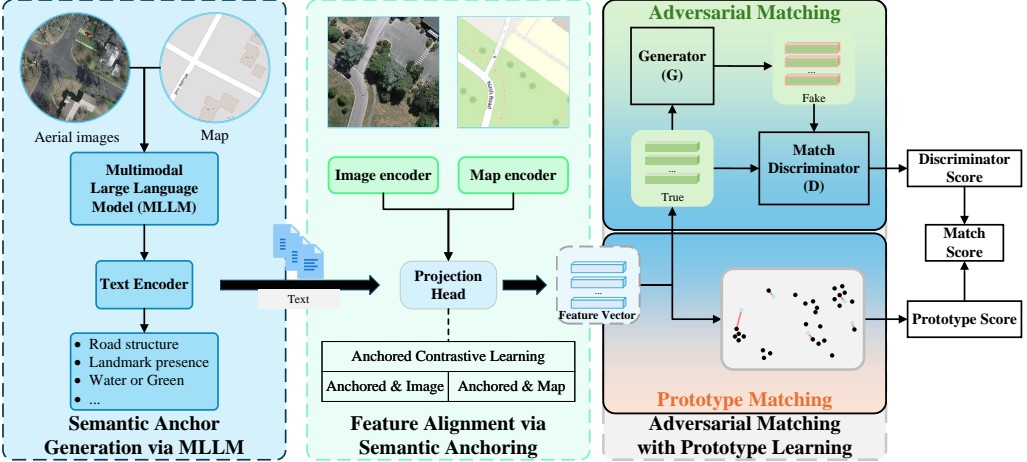

Figure 1: Overview of the proposed U-CSA framework. The process begins with semantic anchor generation via MLLM, followed by feature alignment through anchored contrastive learning, and concludes with adversarial matching enhanced by prototype learning.

## 3 METHOD

We propose a cross-modal image matching framework based on semantic anchoring and adversarial learning, tailored to address the challenge of matching heterogeneous modalities, specifically, aerial imagery and maps. As shown in Figure 1, the core principle is to introduce textual semantics as an intermediate bridge for feature alignment and matching discrimination. The process begins with an MLLM that generates semantically rich text descriptions for image-map pairs, acting as "anchors" connecting the two visual modalities. During the semantic anchoring phase, modality-specific encoders for aerial images and maps are separately optimized via contrastive learning to align their outputs with these textual anchor features within a shared embedding space, thereby establishing implicit semantic consistency. Subsequently, in the adversarial learning phase, a generator-discriminator mechanism is introduced to further refine feature fusion and optimize the matching decision boundary. This is augmented by a learnable prototype library to strengthen both the discriminative power and the interpretability of the model. The final framework provides comprehensive matching confidence scores for unseen image-map pairs, integrating both discriminative and semantic similarity measures. By moving beyond pixel-level correspondence and emphasizing semantic alignment, our approach provides an effective solution for cross-modal matching tasks.

### 3.1 SEMANTIC ANCHOR GENERATION VIA MLLM

The generation of semantic anchors is a key breakthrough in our framework, enabling unsupervised cross-modal semantic alignment. We employ an advanced MLLM (Qwen2.5-VL) as a semantic abstraction engine. Leveraging its deep cross-modal understanding of image-map pairs, the MLLM dynamically extracts high-level, transferable semantic representations —termed "semantic anchors"— which connect the heterogeneous modalities. This process is entirely annotation-free, relying solely on the MLLM's contextual world knowledge and vision-language reasoning capabilities.

MLLM is guided to perform structured visual semantic parsing through carefully designed prompt engineering. The prompts explicitly constrain the model to make inferences based solely on geometric features such as building outlines, spatial topology, and road network structures, while disregarding low-level visual cues like color, texture, and shadows, as well as any textual or Point of Interest (POI) labels. This ensures that the generated semantic descriptions are constrained to a predefined set of discrete semantic attributes, ensuring both cross-modal invariance and a standardized representation of the semantic anchors.

As illustrated in Figure 2, by feeding the MLLM a specific prompt template along with the corresponding image-map pair, the model autonomously outputs a highly structured semantic profile (e.g., in JSON format). This entire process requires no manual intervention or labeled data, leveraging the MLLM's zero-shot cross-modal understanding to achieve automated semantic extraction and alignment, which lays a solid foundation for the subsequent unsupervised matching.

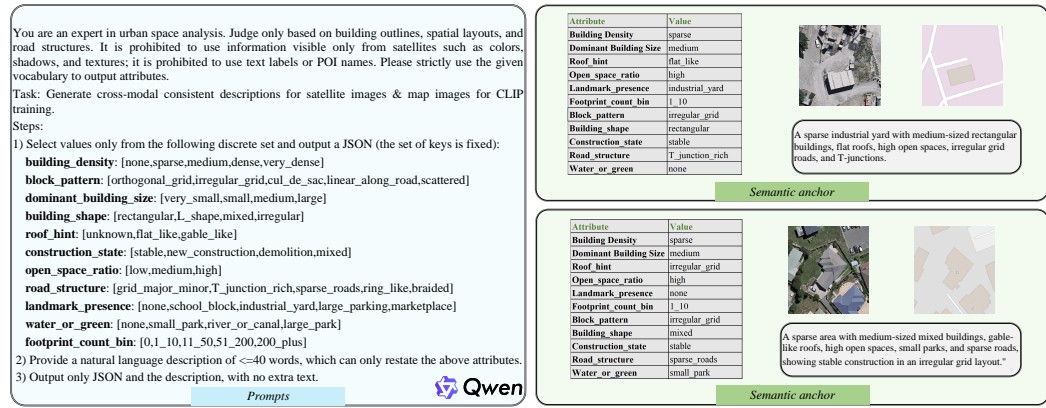

Figure 2: Example of MLLM-based semantic anchor generation. The prompt guides the MLLM (Qwen2.5-VL(Bai et al., 2025)) to focus on structural features and ignore low-level visual cues, producing a structured JSON output along with a concise natural language summary.

## 3.2 FEATURE ALIGNMENT VIA SEMANTIC ANCHORING

Given the high-quality "semantic anchor" descriptions generated for each image-map pair, we introduce Anchored Contrastive Learning—a mechanism that uses semantic anchors as bridges to bring features from aerial images and maps into a unified semantic embedding space. This approach departs from traditional image-image matching paradigms, instead focusing on aligning each modality with its shared semantic anchor.

Our framework consists of a dual-branch vision encoder and a text encoder. The vision encoders process the aerial imagery and maps independently, mapping the visual features into high-dimensional embeddings via an adaptive average pooling layer and a linear projection head. For the text encoder, we use a pre-trained CLIP text Transformer model, keeping its parameters frozen throughout the training process to ensure the stability and consistency of the semantic space. Both visual and text embeddings are projected into the same space to facilitate direct similarity computation.

The core of Anchored Contrastive Learning lies in its novel redefinition of positive pairs: for each image, the positive sample is not a matched image but the corresponding semantic anchor text generated by the MLLM. Assuming a batch of N samples, let the aerial imagery features be $f_i^A$, the map features be $f_i^M$ and the corresponding text features be $f_i^T$ (both L2-normalized). The similarity matrices for image-to-text and map-to-text pairs are calculated as follows:

$$S_{i,j}^{A,T} = \frac{\left(f_i^A\right)^T f_j^T}{\tau}, \quad S_{i,j}^{M,T} = \frac{\left(f_i^M\right)^T f_j^T}{\tau} \tag{1}$$

where $\tau$ is a learnable temperature parameter. The training objective is to minimize a symmetric cross-modal contrastive loss function, which encourages higher similarity for matched image-text pairs os map-text ones and lower similarity for non-matched pairs. The loss is computed for both the image-text and map-text modalities. The symmetric contrastive loss for the image-text pairs is:

$$L_{A\leftrightarrow T} = -\frac{1}{2N} \sum_{i=1}^{N} \left( \log \frac{\exp(S_{i,i}^{A,T})}{\sum_{j=1}^{N} \exp(S_{i,j}^{A,T})} + \log \frac{\exp(S_{i,i}^{A,T})}{\sum_{j=1}^{N} \exp(S_{j,i}^{A,T})} \right) \tag{2}$$

Here, $S_{i,i}^{A,T}$ is the similarity between the $i$-th aerial image and its anchor text, $S_{i,j}^{A,T}$ $(j \neq i)$ is the similarity to non-matching texts, $S_{j,i}^{A,T}$ is the similarity of other images to the $i$-th text. The loss for map-text pairs is similarly defined as:

$$L_{M\leftrightarrow T} = -\frac{1}{2N} \sum_{i=1}^{N} \left( \log \frac{\exp(S_{i,i}^{M,T})}{\sum_{j=1}^{N} \exp(S_{i,j}^{M,T})} + \log \frac{\exp(S_{i,i}^{M,T})}{\sum_{j=1}^{N} \exp(S_{j,i}^{M,T})} \right) \tag{3}$$

During training, only the vision encoders are updated; the text encoder remains frozen. Anchored contrastive learning aligns aerial images and maps with their semantic anchors in a shared embedding space. This ensures that matching image-map pairs have similar feature representations, providing a strong semantic foundation for further matching refinement.

## 3.3 ADVERSARIAL MATCHING WITH PROTOTYPE LEARNING

Building upon the initial semantic alignment of the encoders, we introduce an adversarial matching refinement stage to further improve the discrimination between matched and non-matched pairs, with an emphasis on learning robust non-linear decision boundaries. We adopt a Generative Adversarial Network (GAN) (Goodfellow et al., 2020) framework, where a generator and discriminator jointly optimize feature fusion and matching discrimination through adversarial training.

This architecture comprises four key components: frozen-parameter feature extractors, a feature generator, a discriminator, and a learnable prototype library. The feature extractors are the pre-trained vision encoders from the semantic anchoring stage, which remain fixed and are responsible for extracting high-level semantic features from aerial imagery and maps. The generator fuses and non-linearly maps these heterogeneous features by concatenating the dual-stream outputs and projecting them into a higher-dimensional, discriminative embedding space using a multi-layer perceptron (MLP). The discriminator performs binary classification on these fused features to distinguish between matched and non-matched pairs.

To further enhance the structure and discriminability of feature representations, we introduce a parameterized prototype library module. This module defines a set of optimizable prototype vectors, $P = p_1, p_2, \ldots, p_K$, each representing canonical patterns of semantic alignment in the embedding space. Given a feature representation $f_{gen}$ output by the generator, its similarity to the prototype library is calculated as:

$$S_{proto}(f_{gen}) = \max_{k=1,\ldots,K} \frac{f_{gen} \cdot p_k}{\|f_{gen}\|\|p_k\|} \tag{4}$$

This similarity measure provides a structured, metric-learning-based supervision signal for network learning. Adversarial training employs a dynamic minimax optimization strategy: positive samples correspond to fused features from genuinely matched pairs within a batch, while negative samples are constructed using an in-batch negative sampling strategy, with their generated features denoted as $f_{gen}^{neg}$. The loss function consists of two parts. The first, Discriminator Loss, is optimized with the Hinge loss.

$$L_D = \mathbb{E}[\max(0, 1 - D(f_{gen}))] + \mathbb{E}[\max(0, 1 + D(f_{gen}^{neg}))] \tag{5}$$

where $D(\cdot)$ is the discriminator's output. Prototype Loss is formulated based on the maximum similarity principle:

$$L_{proto} = 1 - S_{proto}(f_{gen}) \tag{6}$$

This Prototype Loss encourages the features of positive samples to align closely with their nearest prototype, leading to compact intra-class distributions. The overall training objective is a weighted combination of these two terms:

$$L_{total} = L_D + \lambda L_{proto} \tag{7}$$

where $\lambda$ is a hyperparameter that balances the contribution of the two losses. The parameters of the generator, discriminator, and prototype library are optimized simultaneously via backpropagation. Through continual adversarial optimization, the model ultimately acquires a more refined and robust capability for discriminating image matching relationships.

### 3.4 Matching Inference

After training, the model evaluates the matching degree of unseen image-map pairs. During inference, the semantic alignment network, first obtains the feature representations for images and maps, which are then fused and transformed into a joint representation vector.

To improve the robustness and interpretability of the matching judgment, the final matching score integrates two complementary mechanisms: Discriminator Score is obtained through the adversarial discrimination of the fused feature:

$$S_{disc} = -D(f_{fused}) \tag{8}$$

where $D$ is the trained discriminator and $f_{fused}$ is the fused feature representation. The score is negated so that a higher value indicates a greater likelihood of a match. Prototype Score measures the semantic relevance between the test pair and learned prototype patterns:

$$S_{proto} = \max_k \text{cosine\_similarity}(f_{fused}, p_k) \tag{9}$$

where $S_{proto}$ is the prototype similarity function. This score reflects the maximum cosine similarity between the test pair and all prototype vectors in the library.

The final matching score is computed as a weighted fusion of the above two scores:

$$S_{final} = \alpha S_{disc} + (1 - \alpha)S_{proto} \tag{10}$$

where $\alpha$ is a hyperparameter controlling the trade-off between the two components. This mechanism yields a quantitative matching confidence for any input pair, where higher scores indicate greater probability of semantic correspondence. Overall, our inference strategy combines discriminative accuracy with semantic interpretability, making it well suited to practical matching decisions in geospatial applications.

## 4 EXPERIMENTS

We evaluate the proposed U-CSA framework on a newly constructed large-scale benchmark, **MST-cons**, designed for reproducible assessment of cross-modal matching between aerial images and maps. This section introduces the dataset, implementation details, evaluation metrics, and comparative methods. We further validate U-CSA through comparisons with state-of-the-art approaches and ablation studies.

### 4.1 DATASETS

To ensure fair and reproducible evaluation, we construct the **Map–Image–Text benchmark (MST-cons)** by integrating high-resolution aerial imagery with corresponding OpenStreetMap (OSM) data(OpenStreetMap contributors, 2017). The aerial imagery comprises two complementary subsets: **MSTcons-WHU**, derived from the WHU Building Change Detection dataset (Ji et al., 2018), and **MSTcons-Inria**, derived from the Inria Aerial Image Labeling dataset (Maggiori et al., 2017) covering five cities. For each aerial image, we retrieve the corresponding OSM map tile such that the image–map pair covers the same geographic region and shares the same spatial extent. All data are further processed into $256 \times 256$ pixel tiles.

Table 1 summarizes dataset statistics. MSTcons-Inria provides 17,347 pairs from Austin, Chicago, Kitsap County, Vienna, and West Tyrol, while MSTcons-WHU contributes 1,560 additional pairs from Christchurch, yielding a total of 18,907 pairs. Explicit train/validation/test splits are maintained to ensure robust evaluation.

Table 1: Per-city dataset statistics (number of image–map pairs).

| Subset | City | Train | Val | Test |
|---|---|---|---|---|
| MSTcons-WHU | Christchurch, New Zealand | 840 | 240 | 480 |
| MSTcons-Inria | Austin, the United States | 2326 | 236 | 472 |
| | Chicago, the United States | 2513 | 718 | 1436 |
| | Kitsap County, the United States | 301 | 86 | 172 |
| | Vienna, Austria | 2954 | 844 | 1688 |
| | West Tyrol, Austria | 1939 | 554 | 1108 |

A distinct feature of MSTcons is the inclusion of *structured semantic anchors*. For each image–map pair, a MLLM generates a JSON object over 11 fixed attributes (e.g., `building_density`, `block_pattern`, `dominant_building_size`, `roof_hint`, `construction_state`, etc.), where each attribute takes values from a predefined discrete vocabulary. In addition, a short natural-language description ($\leq 40$ words) is generated to summarize the JSON attributes. To avoid reliance on superficial cues, the annotator is explicitly restricted from using ephemeral image-only features (e.g., color, shadow, texture) or textual POI labels, and instead must focus on structural information such as building outlines, spatial layouts, and road networks. To ensure data quality, we applied automated we applied automated post-processing and sampling-based manual verification. A random inspection of 500 training pairs yields an overall acceptance rate of 94.2% after reconciliation, with the most common errors arising from roof-hint ambiguities and block-pattern confusions. We release the full MSTcons dataset, including all generated JSON anchors and the inspected subset, to facilitate reproducible comparisons and further research on cross-modal semantic alignment.

### 4.2 IMPLEMENTATION DETAILS

We implement our U-CSA framework using PyTorch. The vision encoders for both aerial images and maps are based on WideResNet-50 architectures, initialized with ImageNet-pretrained weights. The text encoder is a frozen CLIP Transformer. Feature vectors are projected into a 1024-dimensional joint space via a two-layer MLP with ReLU activation. A three-layer discriminator with LeakyReLU activations refines alignment, while a learnable prototype library of 50 vectors provides fine-grained matching cues.

Training is conducted in two stages: (1) *semantic anchoring*, where vision encoders are optimized for 30 epochs, and (2) *adversarial matching*, where the generator, discriminator, and prototype

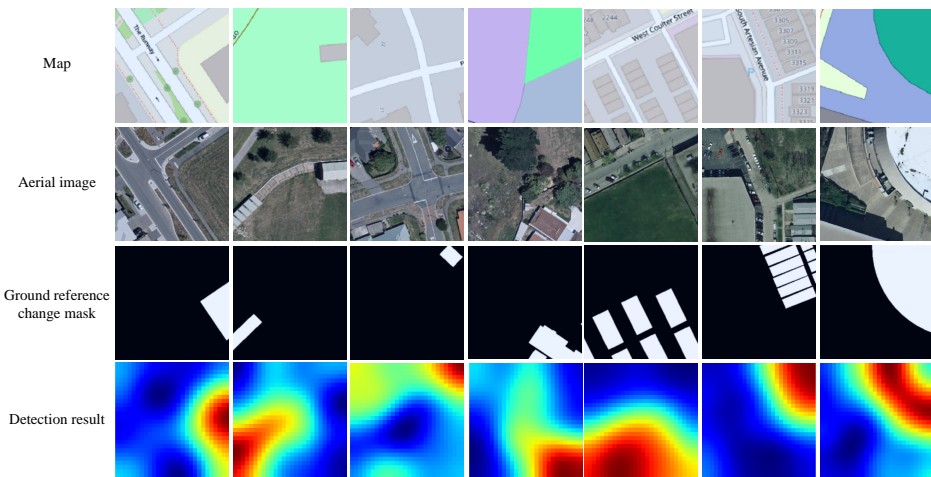

Figure 3: Visualize the detection results: the first row is map, the second row is aerial image, the third row is the ground reference change mask, and the fourth row is the detection result.

library are jointly trained for 20 epochs. Optimization uses Adam ($1 \times 10^{-4}$ for most modules; $5 \times 10^{-5}$ for prototypes), batch size 64, and a learnable contrastive temperature initialized at 0.07. Hyperparameters $\lambda = 0.5$ and $\alpha = 0.7$ are selected via validation. All experiments are run on NVIDIA RTX 5880 GPUs.

### 4.2.1 EVALUATION METRICS

We report two metrics standard in binary matching tasks:

- **ROC-AUC:** threshold-independent discrimination of matched vs. non-matched pairs.
- **F1-score:** harmonic mean of precision and recall, computed at the optimal validation threshold.

Each result is averaged over three independent runs to ensure robustness.

### 4.2.2 COMPARATIVE METHODS

Unsupervised cross-modal matching is relatively unexplored, with SAM-MCD (Chen et al., 2024b) being the only directly related baseline. To provide a comprehensive evaluation, we also compare against:

- **Unsupervised heterogeneous change detection:** methods CDRL(Zhan et al., 2025), CACD(Wu et al., 2021), CFRL(Liu et al., 2025), and PRBCD(Hu et al., 2023), which operate across different remote sensing modalities.
- **Supervised image-map change detection:** methods ObjFormer(Chen et al., 2024a), ChangeFormer (Bandara & Patel, 2022), SEIFNet(Huang et al., 2024), and SNUNet-CD(Fang et al., 2021), adapted to use only matched pairs for consistency with our task formulation.
- **Unsupervised image-map matching:** SAM-MCD, which uses the Segment Anything Model for zero-shot segmentation and change detection between images and maps.

### 4.3 MAIN RESULTS AND ANALYSIS

Figure 3 presents qualitative results of U-CSA. Table 2 compares U-CSA with other methods on MSTcons. U-CSA consistently achieves the highest ROC-AUC and F1-scores across both MSTcons-WHU and MSTcons-Inria subsets, significantly outperforming all baselines. Notably, U-CSA surpasses SAM-MCD by 7.4% in ROC-AUC and 3.3% in F1-score on MSTcons-WHU,

and by 4.9% and 2.7% respectively on MSTcons-Inria. This demonstrates the effectiveness of our semantic anchoring strategy in bridging the modality gap. Although supervised image-map change detection methods demonstrate competitive performance, our unsupervised approach consistently outperforms them. This underscores the advantage of leveraging high-level semantic alignment over pixel-level comparisons. Overall, these results validate the superiority of U-CSA in addressing the challenges of cross-modal semantic alignment and matching. U-CSA can deliver accurate and robust matching scores even in complex urban scenarios by effectively capturing semantic consistency between aerial images and maps.

Table 2: Comparison with state-of-the-art methods on MSTcons.

| Category | SubDataset | MSTcons-WHU | | MSTcons-Inria | |
|---|---|---|---|---|---|
| | METHOD | AUROC | F1 | AUROC | F1 |
| Unsupervised heterogeneous change detection | CDRL | 55.2 | 58.5 | 52.1 | 50.5 |
| | CACD | 51.1 | 35.9 | 49.4 | 41.1 |
| | CFRL | 54.1 | 51.1 | 46.8 | 40.2 |
| | PRBCD | 55.2 | 51.1 | 45.4 | 39.1 |
| Supervised image-map change detection | ObjFormer | 60.5 | 57.6 | 47.7 | 55.6 |
| | ChangeFormer | 52.7 | 51.0 | 44.5 | 50.1 |
| | SEIFNet | 58.5 | 65.7 | 57.7 | 54.1 |
| | SNUNet-CD | 58.9 | 56.1 | 54.1 | 51.2 |
| Unsupervised image-map | SAM-MCD | 71.1 | 67.5 | 64.5 | 58.7 |
| - | **U-CSA** | **78.5** | **70.8** | **69.4** | **61.4** |

Table 3: Ablation study results on MSTcons. Each row represents a different ablation configuration. The best results are highlighted in bold. SA = Semantic Anchor, CL = Contrastive Learning, PL = Prototype Learning.

| SA | CL | PL | MSTcons-WHU | | MSTcons-Inria | |
|---|---|---|---|---|---|---|
| | | | ROC-AUC | F1 | ROC-AUC | F1 |
| × | Image-Map | ✓ | 72.3 | 64.5 | 64.7 | 57.8 |
| ✓ | Anchored | × | 76.5 | 68.1 | 67.8 | 59.9 |
| ✓ | Anchored | ✓ | **78.5** | **70.8** | **69.4** | **61.4** |

## 4.4 ABLATION STUDIES

To further validate the effectiveness of each component within the U-CSA framework, we conduct comprehensive ablation studies on the MSTcons benchmark. In these experiments, we systematically remove or replace key modules and observe the impact on model performance. The results are summarized in Table 3. The results indicate that removing the semantic anchor generation —by replacing it with direct image-map contrastive learning—significantly degrades performance, highlighting the critical role of high-level semantic guidance. Additionally, omitting the prototype learning module during the adversarial matching stage results in lower accuracy, demonstrating its importance for feature discriminability. Collectively, these findings confirm that each component is integral to the overall effectiveness of U-CSA, thereby validating our architectural choices.

## 5 CONCLUSION

In this paper, we introduced U-CSA, an innovative framework for unsupervised cross-modal semantic alignment and matching between aerial imagery and maps. Unlike conventional methods that depend on low-level visual features, U-CSA leverages semantic anchors generated by MLLMs to establish a unified semantic space. A dual-branch encoder with adversarial refinement further enhances discriminative capability. Experiments on the newly constructed MSTcons benchmark demonstrate the superiority of U-CSA over state-of-the-art methods. Beyond quantitative gains, U-CSA illustrates how MLLMs can serve as scalable annotators, producing interpretable and domain-specific anchors. Future work will explore anchor reliability, additional modalities (e.g., LiDAR, street-view), and temporal dynamics for continuous urban monitoring.

## 6 ETHICS STATEMENT

This work does not involve human subjects, personally identifiable information, or sensitive data. All datasets used are publicly available and widely adopted in the research community. Our methods are developed solely for academic research purposes, without any intent of misuse or deployment in harmful applications. We have carefully considered issues of fairness, privacy, and potential societal impact, and we believe our study poses no ethical concerns.

## 7 REPRODUCIBILITY STATEMENT

We provide all necessary details, including model architectures, training procedures, and hyperparameters, to facilitate reproducibility. The MSTcons dataset, along with the generated JSON anchors and the inspected subset, will be made publicly available to support further research and validation of our findings.

## 8 THE USE OF LARGE LANGUAGE MODELS

We used large language models (LLMs) exclusively for improving the clarity, readability, and professionalism of the manuscript text. No LLMs were employed in data collection, data processing, analysis, or experimental result generation.

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
