# OpenReview forum: "Cross-Modal Semantic Anchoring: Unsupervised Consistency Verification for Aerial Imagery and Maps via Multimodal LLMs"
_ICLR.cc/2026/Conference — ICLR 2026 Conference Withdrawn Submission_

### Official Review · Reviewer_5nHU · 2025-10-26

**Soundness:** 2
**Presentation:** 2
**Contribution:** 2
**Rating:** 4
**Confidence:** 4

**Summary:**

The paper tackles cross-modal matching between aerial imagery and maps. It uses a multimodal LLM (e.g., Qwen2.5-VL) to generate semantic anchors, a structured JSON plus a short textual summary describing salient topology (roads, rivers, blocks). Image and map encoders are trained via anchor-guided contrastive learning to align to the same anchor space; a prototype-augmented, adversarial matching head then outputs a match confidence. A new dataset of image–map–text triplets is introduced, and results are reported mainly with ROC-AUC/F1 on a binary “match vs. non-match” task.

**Strengths:**

1. Using text/JSON anchors to reduce the appearance gap between photos and symbolic maps is conceptually neat and interpretable.
2. No human pixel annotations are required; the framework can operate when true negatives are scarce, which is common in matchingtasks.
3. The combination of anchor contrastive learning + prototype bank + adversarial matcher is coherent, and ablations suggest each piece contributes.

**Weaknesses:**

1. The paper motivates precise map updating, but the core task and metrics are global matching (AUC/F1). There is no quantitative localization (e.g., object/pixel-level) to support “fine-grained updates.” Consider adding a quantitative link from match scores to updatable regions.
2. Many baselines are change-detection methods designed to output segmentation masks; adapting them to a binary matching regime may undercut their strengths. Add more matching-oriented baselines to strengthen fairness.
3. Performance hinges on MLLM-generated anchors. The paper lacks sensitivity studies across different MLLMs/prompts, and does not quantify how anchor noise affects final accuracy.
4. It is unclear how non-match pairs are formed and whether geographic leakage exists. Without clear cross-city/region leave-one-out splits, AUC may be optimistic or pessimistic depending on how “easy” the negatives are.
5. Beyond AUC/F1, include PR curves, calibration, confidence intervals, and threshold-selection policy. If map updating is a goal, add object/pixel metrics (IoU/mAP) on a subset, or report retrieval metrics for “find the correct map tile” scenarios.
6. Anchor generation with an MLLM can dominate cost. Please quantify end-to-end training/inference time, GPU memory, and anchor generation throughput.
7.

**Questions:**

1. How are non-match pairs constructed at train/test time? Do you provide cross-city/country splits to avoid geographic leakage?
2. How sensitive is the method to the choice of MLLM and prompt design? What happens if you swap in a weaker/stronger model or perturb prompts?
3. Can the system produce localization (e.g., building/road-level masks or vector changes) or a proxy that correlates with localization metrics (IoU/mAP)?
4. Could you report native-setting results for change-detection baselines as an upper bound, and add more matching-specific baselines to ensure fairness?
5. What are the compute/latency numbers for anchor generation and matching on a standard GPU? Any plan for anchor caching or text-space distillation to reduce cost?
6. What is the contribution of the JSON vs. free-text parts of the anchor, and how does prototype count K affect accuracy/calibration and interpretability?

---

### Official Review · Reviewer_WjuD · 2025-10-26

**Soundness:** 3
**Presentation:** 2
**Contribution:** 2
**Rating:** 4
**Confidence:** 4

**Summary:**

In this paper, the author argues that the text, as semantic guidance, to supervise the maps as well as the aerial images.

My main concern is why not directly using a finetuned CLIP or BLIP model. I do not see the technical difference on semantic guidence.

The inference process is not very clear.  I guess that the author uses the matching score to detect the building / infrastructure changes. More detail illustrations are required.

The setting is ambiguous. The proposed method still needs paired image-map to generate the anchor text to train the model.

The total loss does not include the contrastive loss in Eq.2 and Eq.3.

T is ambigous, some time denotes Text, sometime is vector transform (see Eq.1).

**Strengths:**

In this paper, the author argues that the text, as semantic guidance, to supervise the maps as well as the aerial images.

1. Leverages multimodal large language models (MLLMs) to generate interpretable semantic anchors, improving cross-modal alignment of aerial imagery and maps beyond low-level feature matching.

2. Exploits MLLMs’ world knowledge, reducing reliance on large paired datasets compared to traditional contrastive pre-training methods.

3. With improved alignment accuracy over state-of-the-art methods in urban map updating tasks.

**Weaknesses:**

1. Ambiguous Training Setup: Claims unsupervised learning but seems to require paired image-map data for anchor text generation, undermining the unsupervised claim.

2. Unclear Inference Process: Lacks detailed explanation of how matching scores detect urban changes, limiting practical understanding.

3. Incomplete Loss Formulation: Total loss omits contrastive loss terms (Eq. 2, Eq. 3), creating confusion about optimization.

4. Notational Inconsistency: Variable $ T $ is ambiguously used for text and vector transformation, reducing clarity.

**Questions:**

1. Novelty: How does the proposed method’s semantic anchoring technically differ from or improve upon finetuned CLIP/SigLIP/BLIP models for cross-modal alignment? How about the performance that we directly finetune CLIP/SigLIP/BLIP models with image-map matching?

2. Can the authors clarify the inference process, particularly how matching scores are used for detecting building/infrastructure changes, with illustrative examples?

3. Does the training process truly avoid paired data, or are image-map pairs required for anchor text generation?

4. Why are the contrastive loss terms (Eq. 2, Eq. 3) excluded from the total loss, and how does this affect training?

5. Can the authors resolve the notational ambiguity of $ T $ to improve the paper’s clarity?

---

### Official Review · Reviewer_tB41 · 2025-10-28

**Soundness:** 3
**Presentation:** 2
**Contribution:** 2
**Rating:** 4
**Confidence:** 4

**Summary:**

This paper focuses on cross-modal matching between aerial imagery and abstract maps. It uses semantic anchors generated by a multimodal large model as intermediaries to align the two modalities into a shared semantic space (anchor-based contrastive learning), and then refines the match decision via adversarial matching plus a prototype library. The authors also construct the MSTcons benchmark to enable reproducible experiments. Core results show that the proposed U-CSA yields significant AUROC and F1 gains over both unsupervised and supervised baselines.

**Strengths:**

1.The paper explores using large models to generate text, which aligns with current research trends; compared with unsupervised and supervised baselines, the proposed U-CSA achieves significant gains in AUROC and F1.

2.Ablation studies show that removing the semantic anchors or the prototype learning module results in a significant performance drop.

3.The work introduces MSTcons (18,907 pairs) with clear city-level statistics and explicit train/validation/test splits.

**Weaknesses:**

1.Reliance on MLLM-generated anchors may introduce bias and inconsistency; the authors also list “anchor reliability” as future work, indicating remaining uncertainty.

2.A random spot-check of 500 pairs shows a 94.2% acceptance rate, with common errors such as confusion between rooftops and block patterns, implying noisy training signals; the sensitivity of performance to this noise is not reported.

3.Several supervised baselines were constrained to use only matched pairs to fit the paper’s task setup, which may systematically undercut their strengths and compromise fairness.

4.Semantic anchors are generated offline by an MLLM, but generation latency and compute/token costs are not reported.

5.Only OSM is used, with tiles uniformly cropped to 256×256; generalization to other cartographic styles/zoom levels or larger spatial contexts is unverified.

6.The prototype library and the fusion weights lack more extensive ablation studies.

**Questions:**

1.	The “LLM usage statement” claims no LLMs were used for data collection/processing/experimental outputs, yet the methodology explicitly employs an MLLM to generate semantic anchors; this is contradictory, please clarify.

2.	Can comparable results be achieved when replacing the current MLLM with alternative MLLMs?

3.	Please refine the figures and tables to be more aesthetically polished and consistent.

---

### Official Review · Reviewer_iDAK · 2025-10-29

**Soundness:** 2
**Presentation:** 2
**Contribution:** 2
**Rating:** 2
**Confidence:** 4

**Summary:**

The paper proposes U-CSA—an unsupervised cross-modal semantic anchoring framework to match aerial imagery with vector maps. Instead of aligning image↔image, U-CSA first asks a multimodal LLM (Qwen2.5-VL) to produce structured “semantic anchors” (JSON over 11 attributes + a ≤40-word summary) for each image–map pair; then (i) a dual-branch visual encoder is trained with anchored contrastive learning against the text anchors; and (ii) an adversarial matching head with a prototype library refines the decision boundary. The authors also introduce MSTcons, a 18,907-pair benchmark built from WHU (Christchurch) and Inria (Austin, Chicago, Kitsap, Vienna, West Tyrol), with 256×256 tiles and explicit splits. On MSTcons, U-CSA beats unsupervised SAM-MCD and several adapted change-detection baselines in ROC-AUC/F1, with ablations supporting the contribution of anchors and prototypes.

**Strengths:**

Originality. Recasts image↔map matching as visual↔text (anchor) alignment, using MLLM-generated semantic anchors to bridge a large modality gap—moving the objective from pixel similarity to semantic consistency.
Quality. A clear two-stage training (anchored contrastive → adversarial+prototypes), with formulas and losses spelled out, plus ablations showing each component matters.
Clarity. Good problem framing and Figure 1 overview; prompt template and anchor examples (Figure 2) make the pipeline interpretable.  Significance. Demonstrates that anchor-driven semantic alignment can outperform pixel-level or zero-shot segmentation baselines for unsupervised image–map matching on two cities-level subsets and overall.

**Weaknesses:**

LLM-usage inconsistency / auditing. Section 3.1 relies on Qwen2.5-VL to generate anchors and even dataset attributes, yet “Use of LLMs” claims LLMs were used only for manuscript editing. Please reconcile and audit anchor generation (versions, prompts, seeds, temperature), since the anchorer is effectively an external supervision source.
Anchor reliability & bias. You report a 94.2% acceptance on 500 sampled pairs with common errors in roof-hint and block-pattern. Provide per-attribute accuracy/IAA, confusion matrices, and how anchor noise propagates to training (robust loss? filtering? multiple anchorers?). Compare different MLLMs / prompt sets to bound sensitivity.
“Unsupervised” claim clarity. The method uses MLLM world knowledge and in-batch negatives in the adversarial stage. Please position the work as no human labels for match/non-match, and quantify how much gain comes specifically from text anchors vs. (a) direct image–map InfoNCE, (b) CLIP-style hand-crafted prompts (no MLLM), (c) anchors from image-only or map-only inputs.
Evaluation breadth. MSTcons covers 6 cities; add leave-one-city-out generalization and a different continent/provider to test robustness. Also report calibration (AURC/ECE) and retrieval settings (Top-K among distractor maps/images), which better reflect practical updating.

**Questions:**

LLM disclosure. Which Qwen2.5-VL checkpoint, decoding params, and prompt templates were used? Can you release all prompts and a minimal anchor-generator to reproduce anchors deterministically?
Anchor reliability. What is per-attribute IAA/accuracy and how do anchor errors affect downstream ROC-AUC/F1? Any confidence-weighted training using JSON agreement or multiple anchorers?
Generalization. How does U-CSA perform cross-city (train on A,C,K,V,W; test on Christchurch) and vice versa? Any degradation when anchors are generated by a different MLLM?
Ablation scope. Can you add: (a) no-anchor InfoNCE; (b) hand-crafted textual prompts as anchors; (c) map-only vs image-only anchors; (d) negative-sampling strategies beyond in-batch?
Practical metrics. Will you report retrieval Top-K among many candidates and calibration (ECE/Brier) to reflect deployment needs for automated map updating?

---

### Note · Authors · 2025-11-25

I have read and agree with the venue's withdrawal policy on behalf of myself and my co-authors.